# Salivary Carbohydrate-Deficient Transferrin in Alcohol- and Nicotine-Dependent Males

**DOI:** 10.3390/jcm9124054

**Published:** 2020-12-15

**Authors:** Napoleon Waszkiewicz, Katarzyna Pawłowicz, Natalia Okuniewska, Mikołaj Kwiatkowski, Daniel Zalewski, Karolina Wilczyńska, Agata Szulc, Beata Galińska-Skok, Beata Konarzewska, Mateusz Maciejczyk, Anna Zalewska

**Affiliations:** 1Department of Psychiatry, Medical University of Białystok, Plac Brodowicza 1 Str., 16-070 Choroszcz, Poland; katarzyna.pawlowicz89@gmail.com (K.P.); nkokuniewska@gmail.com (N.O.); hejjj@wp.pl (M.K.); dnlzalewski@gmail.com (D.Z.); karolinawilczynska88@gmail.com (K.W.); begal@poczta.onet.pl (B.G.-S.); beatajan@op.pl (B.K.); 2Department of Psychiatry, Medical University of Warsaw, Partyzantów 2/4 Str., 05-802 Pruszków, Poland; agataszulc@poczta.onet.pl; 3Department of Experimental Pharmacology, Medical University of Bialystok, Szpitalna 37 Str., 15-295 Bialystok, Poland; mat.maciejczyk@gmail.com; 4Experimental Dentistry Laboratory, Medical University of Bialystok, 15-222 Bialystok, Poland; azalewska426@gmail.com

**Keywords:** carbohydrate-deficient transferrin, alcohol, nicotine, markers, saliva

## Abstract

Serum carbohydrate-deficient transferrin (CDT), an 80 kDa glycoprotein, is one of the most commonly employed biomarkers to detect alcohol dependence. Some salivary glycoproteins such as α-amylase, clusterin, haptoglobin, light/heavy-chain immunoglobulin, and transferrin, which alter glycosylation in alcohol-dependent persons, have been suggested to be potential alcohol markers. However, their identification is based on indirect analysis of lectin glycosidic bonds and molecular weight. We investigated the CDT content in the saliva of alcohol- and nicotine-dependent men. The CDT concentration (ng/mL, ng/mg protein) was determined by an Enzyme-Linked Immunosorbent Assay (ELISA) commercial kit in 55 men: 20 healthy social drinkers (C), 10 chronic cigarette smokers (S), 10 alcohol-dependent non-smokers (A), and 15 alcohol-dependent smokers (AS). Surprisingly, there were no differences in the concentrations of CDT between the studied groups. Salivary pH was the lowest in the AS and the highest in the A group. Therefore, salivary CDT cannot be used as an alcohol dependence marker as measured by ELISA. We suggest that direct identification of glycoproteins is necessary to search for potential salivary alcohol biomarkers. Molecules smaller than 40 kDa, which easily translocate from blood to the saliva, might be preferred as salivary alcohol markers.

## 1. Introduction

Alcohol dependence affects from 2% to 4% of the world population, of which 80% to 90% smoke cigarettes [1,2,3,4,5]. Salivary diagnostics in addictions is increasing, as saliva contains many compounds that sensitively respond to toxic substances, which reflects real-time levels of these compounds/markers. Saliva collection is non-invasive, stressless, poses no risk of tissue injury, and is easy to self-perform [6,7,8,9,10,11]. Some potential salivary markers of chronic ethanol use have been proposed so far, including aminotransferases, gamma-glutamyl-transferase [12], ethanol [13,14], sialic acid [15], hexosaminidase A, glucuronidase [9], some ethanol poisoning congeners such as methanol [16], etc. An earlier study found significant differences in glycosylation (α2,3-sialylation, fucosylation, and expression of T-antigen) of salivary α-amylase, clusterin, haptoglobin, light and heavy chains of immunoglobulin, and transferrin between alcohol-dependent and healthy persons [17]. These new identified salivary glycoproteins were suggested as potential markers to diagnose alcohol dependence syndrome. However, their identification was based on an indirect method, through the analysis of lectin glycosidic bonds and molecular weight of glycoproteins [17]. 

Carbohydrate-deficient transferrin (CDT) is the only test that has been approved by the Food and Drug Administration in the identification of heavy alcohol use. It is a glycoform profile of serum transferrin that is hyposialylated and appears in serum after regular high alcohol intake (more than 50–80 g of alcohol/day for at least 1 week). CDT is a glycoprotein that has been widely investigated as the most specific of the available methods to detect chronic alcohol drinking/dependence; the sensitivity and specificity in alcohol-dependent persons are high: −82% and 97%, respectively. During abstinence, CDT normalizes with a half-life of 15 days and can remain elevated for several weeks. If drinking resumes, even low levels of ethanol can result in its rapid re-elevation [18,19,20].

As salivary transferrin is suggested to be potential marker of alcohol dependence or chronic alcohol drinking, we investigated if changes in transferrin glycosylation would result in a real formation of the salivary alcohol marker CDT. To exclude the effect of cigarette smoking on CDT values, we measured CDT in controls, smoking social drinkers, and in smoking and non-smoking alcohol-dependent persons.

## 2. Methods

### 2.1. Participants

Fifty-five men were recruited to the study: 20 healthy social drinkers (C group) without a history of harmful alcohol use/dependence or smoking, 10 chronic cigarette smokers (social drinkers; drink not more than 20–30 g of pure ethanol per day; S), 10 alcohol-dependent non-smokers (A; inpatients), and 15 alcohol-dependent smokers (AS; inpatients). Table 1 shows the descriptive statistics for the studied groups. A and AS group participants were recruited from the Treatment of Alcohol Withdrawal Symptoms unit in Choroszcz, Poland, and salivary samples were collected on the 1st day of the abstinence period (after chronic alcohol consumption). Samples from the C and S groups were collected from volunteers. Alcohol and nicotine dependence were diagnosed according to the International Classification of Diseases 10th Revision (ICD-10) and the Diagnostic and Statistical Manual of Mental Disorders IV Edition (DSM-IV) criteria. Groups C and S involved volunteers that were recruited from the Choroszcz Hospital Staff. The number of cigarettes smoked per day, years of nicotine dependence, as well as the number of days and grams of daily alcohol consumed and years of alcohol dependence were determined from interviews with the patients/volunteers and by a professional psychiatrist with extensive clinical experience. The cut-off for chronic alcohol drinking (in alcohol-dependent persons—A and AS groups) was at least 3 consecutive days of drinking at least 100 g of pure ethanol per day. Participants from the A and AS groups suffered from alcohol withdrawal. The cut-off for social alcohol drinking (C and S groups) was consuming not more than 20–30 g of pure ethanol per day. Volunteers had not consumed alcohol for 10 days before study recruitment, and ”0” alcohol data were provided for the last alcohol consumption (days) and the daily amount of the last alcohol consumption (g).

### 2.2. Ethical Issues

The study was approved by the local Bioethical Committee of Medical University of Białystok and conducted in accordance with the Helsinki Declaration. Informed written consent was obtained from all subjects after explaining the nature, purpose, and potential risks of the study. 

### 2.3. Procedures

#### 2.3.1. Data and Sample Collection 

All samples (3 mL) of unstimulated (residual) whole saliva were collected into plastic tubes on ice by spitting, under standardized conditions [21,22], between 8:00 and 9:00 am to minimize the influence of circadian rhythms. Samples were centrifuged (3000× *g*; 20 min; 4 °C) to remove cells and debris. The supernatants in 200 μL portions were frozen and kept at −80 °C until analysis.

#### 2.3.2. Analytical Methods

The CDT concentration in the saliva was determined in duplicate samples using a commercial Enzyme-Linked Immunosorbent Assay (ELISA) kit (Human CDT (Carbohydrate Deficient Transferrin) ELISA Kit, Elabscience Biotechnology Co. Ltd., Wuhan, China). According to the manufacturer’s instructions, supernatants of saliva samples were incubated with an antibody specific for CDT and then with a secondary antibody labeled with Avidin Horseradish Peroxidase (HRP). After incubation with the substrate, the enzyme substrate reaction was terminated by sulfuric acid, and the color change was measured at 450 nm by a microplate reader, Mindray MR-96. The sensitivity of the assay was 9.38 ng/mL and the detection range was 15.63–1000 ng/mL. This kit recognizes human CDT in samples. No significant cross-reactivity or interference between human CDT and analogues was observed (https://www.elabscience.com/PDF/Cate61/E-EL-H0638-Elabscience.pdf).

The salivary protein content (Sp) was determined using the bicinchoninic acid method (BCA) (BCA Protein Assay Kit; PIERCE, Rockford, IL, USA), with bovine albumin as the standard. All analyses were performed in duplicate.

### 2.4. Statistical Analysis

Statistical analysis was performed with Statistica version 13 (Statsoft, Cracov, Poland). All presented data were tested for normal distribution, and nonparametric methods (most appropriate for the small sample sizes) were used. Then, results were expressed as medians (CI; confidence interval). Comparisons between parameters in C, S, A, and AS groups were analyzed using the Kruskal–Wallis test, and differences were measured using the Mann–Whitney “U” test. Spearman’s rank correlation coefficient was used to measure the statistical link between two variables. Statistical significance was defined at *p* < 0.05.

## 3. Results

### 3.1. Descriptive Characteristics of Studied Groups

There were no statistically significant differences in age and body mass index (BMI) between C, S, A, and AS groups (Table 1). There were also no statistically significant differences in duration (in days) and daily amount (in grams) of the last alcohol consumption between A and AS groups. The AS group smoked cigarettes for significantly longer and in higher numbers than group S did (Table 1).

### 3.2. Salivary Proteins and CDT

We found differences in salivary pH between the studied groups, with the highest pH in group A and the lowest pH in group AS (Table 1). We did not find statistically significant differences in salivary protein content nor CDT (in ng/mL and ng/mg of protein) between the groups (Kruskal–Wallis analysis; Table 1; Figure 1). 

Correlations were found between salivary pH and CDT (ng/mL) in the control group (r = 0.540, *p* = 0.010), as well as between CDT (ng/mL) and daily amount of the last alcohol consumption (g) (r = 0.615, *p* = 0.025) and between CDT (ng/mg of proteins) and the length of alcohol consumption (in days) (r = 0.671, *p* = 0.016) in the AS group.

## 4. Discussion

Ethanol and cigarette smoking, through their various metabolites such as acetaldehyde and reactive oxygen species (ROS), can destroy tissues of the human body, including oral tissues with salivary glands [4,5,9,23,24,25,26,27,28]. Ethanol diffuses rapidly into the saliva and oral tissues, and immediately after drinking, the salivary concentration is temporarily much higher than in plasma. At the same time, the level of the ethanol metabolite acetaldehyde in the saliva is 10 to 100 times higher than the level in the blood [9,29,30]. Acetaldehyde in the oral cavity of smoking alcohol-dependent persons comes from ingested ethanol and tobacco smoke. Besides acetaldehyde, tobacco smoke is a source of oxidative stress and contains up to 3000 toxic substances, such as nicotine, carbon monoxide, nitrosamines and other aldehydes, which may damage the oral tissues [9,31]. Moreover, ROS generated during drinking and smoking, as well as non-oxidative metabolites of ethanol such as fatty acid ethyl esters (FAEEs) and the ethanol/water competition mechanism, might be involved in the resulting tissue damage [9,19,32].

Except for some identified salivary alcohol markers that were observed to increase from chronic drinking such as aminotransferases, gamma-glutamyl-transferase, ethanol, sialic acid, glucuronidase, hexosaminidase A, and methanol (ethanol congener), some other salivary glycoproteins such as α-amylase, clusterin, haptoglobin, light and heavy chain immunoglobulin, and transferrin were suggested to be worth further detailed examination [9]. A pilot study led by Kratz [17] tried to identify some salivary glycoproteins that were altered by chronic alcohol consumption. By using defined molecular weights of glycoproteins and glycosidic bonds, they identified salivary glycoproteins with significant alcohol-induced differences in glycosylation (α2,3-sialylation, fucosylation, and expression of T-antigen). These salivary glycoproteins, changed by chronic alcohol use, included α-amylase, clusterin, haptoglobin, light and heavy chain immunoglobulin, and transferrin. Therefore, it was suggested that these salivary glycoproteins might be potential markers to diagnose alcohol dependence syndrome. 

We, surprisingly, discovered that chronic alcohol use did not increase CDT levels in saliva (Table 1; Figure 1). Therefore, salivary CDT measured by ELISA seems not to be suitable for the diagnosis of alcohol dependence syndrome. We also found no statistical differences in salivary protein content nor in CDT/protein levels between C, S, A, and AS groups. However, the AS group had the lowest and the C group the highest CDT levels. It may be that CDT does not pass from the blood to the saliva, even if oral cavity tissues are injured by alcohol and smoking. Human transferrin is a glycoprotein with a molecular weight of 80 kDa and has two iron-binding sites [33]. The transfer of a substance from blood to the saliva seems to be dependent on its lipid solubility, pKa, plasma protein binding, and on its molecular weight [34,35]. Generally, unionized substances with high lipid solubility and low molecular weight are more likely to pass to the saliva from the blood. It was found that approximately 27% of whole salivary proteins are also found in the plasma [35]. The highest fraction of proteins found in the whole (mixed) saliva range in size between 20 and 40 kDa, whereas the 40 to 60 kDa range is the largest fraction for plasma [35]. Therefore, the molecular weight of CDT (even without sialic acid residues) may still be too high to translocate from blood to the saliva. Since the average pH of saliva (6.5) is lower than the average pH of plasma (7.4), and saliva contains more proteins in the lower acidic end (isoelectric point pI ≤ 5; 10.5% for saliva compared with 6.7% for plasma) [35], it is possible that CDT without sialic acid chains may be too weak an acid to translocate to the saliva. In the AS group, which had the lowest CDT concentration, salivary pH was also the lowest, which might also be due to the synergistic action of alcohol consumption and smoking. We also found correlation between salivary pH and CDT levels. However, salivary pH was lowest in the AS group and highest in the A group. Smoking alcohol-dependent persons (AS) had the longest time of alcohol dependence and the longest time and the largest amount of the cigarette smoking. Therefore, salivary CDT might be potentially destroyed by the combined action of alcohol and cigarette smoke and by their metabolites. On the other hand, it may also be that differences in glycosylation, identified in Kratz et al.’s study [17], were found in alcohol-dependent persons that covered smokers and non-smokers in one alcohol group. Hence, these differences in glycosylation likely are due to the synergistic action of alcohol and cigarette smoke. In our study, alcohol-dependent smoking persons had the lowest CDT values. We might also mention that the marker’s identification, based on differences in glycosylation and through indirect analysis of lectin glycosidic bonds and molecular weight of glycoproteins, may be misleading.

Some limitations of our present study include the small group sizes and absence of oral cavity parameters (oral hygiene, dental and periodontal state). Therefore, results need to be replicated in further studies based on a larger population with a detailed oral cavity check. As the best method for chronic alcohol use is the measurement of %CDT by high-performance liquid chromatography (HPLC) with UV detection [20,36], a simple ELISA commercial kit, as in our study, may result in worse sensitivity and specificity. Another limitation may be due to the fact that the number of cigarettes smoked per day and the number of daily drinks were determined only from interviews with the patients and volunteers. Although no significant cross-reactivity or interference between human CDT and analogues was described in the manufacturer’s manual and datasheet, another limitation may be due to the potential cross-reactivity of ELISA with other proteins in the saliva [37]. Unfortunately, the glycosylation profile of the measured transferrin was not described in our study, which is also another study limitation. Since pH variations may have significant effects on protein antigens (depending on their charge) and may chemically change the antibody itself [38], the results obtained from our assays may not reflect the true functional affinity of antibodies to CDT, and this may also be a limitation of our study. However, although the AS group had the lowest salivary CDT concentration and pH, we found a correlation between salivary pH and CDT only in the control group. As smoking can modify CDT levels by inducing hepatic microsomal enzyme activity [39], and smoking particularly damages the oral tissues [5], we cannot exclude the influence of smoking on salivary CDT in our study. The link between serum CDT levels and factors such as age, sex, body mass index, ethnicity, and smoking was noted as small and considered clinically insignificant [20]. Therefore, the adjustment of CDT for these factors is generally not required [20]. Moreover, the effect of cigarette smoking on CDT was excluded in our study by recruiting various groups, including smoking and non-smoking controls and alcohol-dependent persons, which is the strength of our study.

## 5. Conclusions

In conclusion, we show, for the first time, that salivary CDT cannot be used as an alcohol dependence marker when measured by an ELISA commercial kit. Other salivary glycoproteins, such as α-amylase, clusterin, haptoglobin, and light and heavy chain immunoglobulin, identified as potential markers of alcohol dependence need further direct research, based on a larger sample. Indirect identification of glycoproteins, based on differences in glycosylation, may not be a suitable method of searching for salivary biomarkers in smoking alcohol-dependent persons. Not only smoking persons but also non-smoking alcohol-dependent persons need to be recruited in salivary alcohol marker studies. Molecules smaller than 40 kDa that can easily translocate from blood to the saliva might be preferred as salivary alcohol markers.

## Figures and Tables

**Figure 1 jcm-09-04054-f001:**
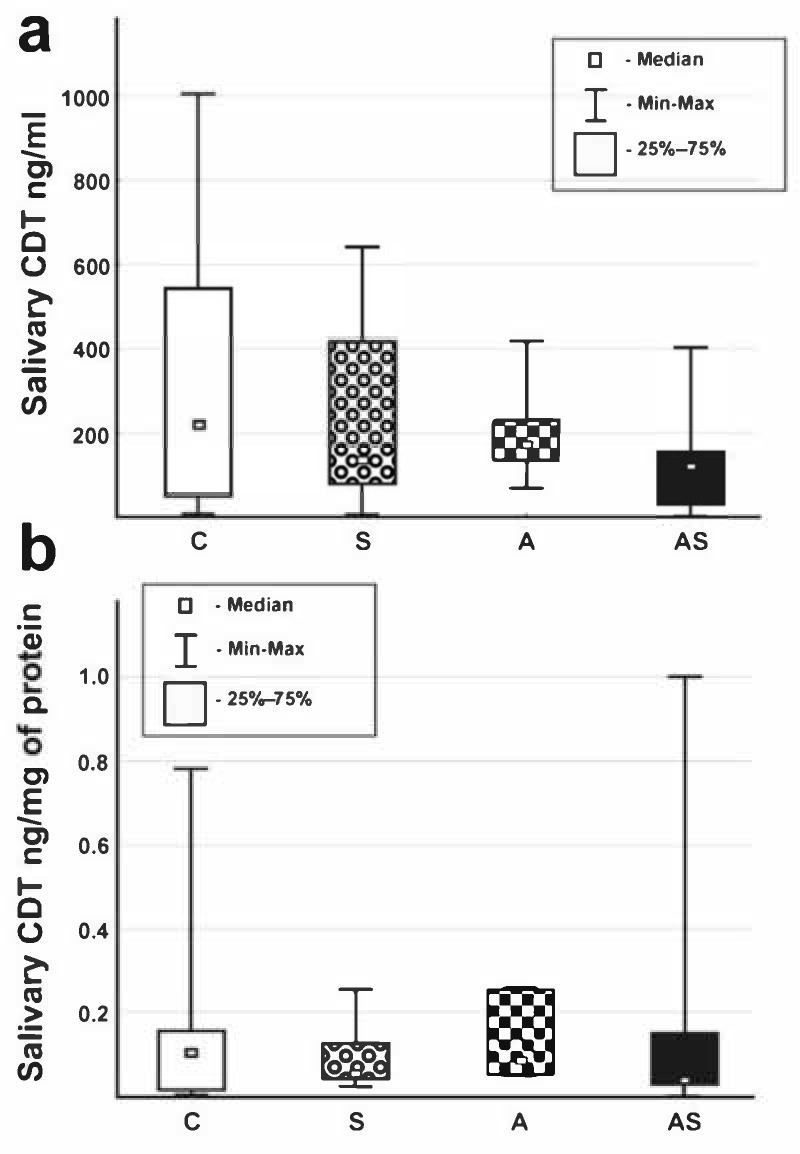
The salivary carbohydrate-deficient-transferrin (CDT) content ((**a**) ng/mL of saliva, (**b**) ng/mg of protein) in healthy social drinkers (C group), chronic cigarette smokers (without harmful alcohol use/dependence; S), alcohol-dependent non-smokers (A), and alcohol-dependent smokers (AS). No significant differences were observed in the Kruskal–Wallis analyses.

**Table 1 jcm-09-04054-t001:** Group characteristics and salivary parameters (pH, protein content, carbohydrate-deficient transferrin (CDT)) in healthy social drinkers (C group), chronic cigarette smokers (without harmful alcohol use/dependence; S), alcohol-dependent non-smokers (A), and alcohol-dependent smokers (AS).

Variable (Concentration)	C (Min–Max) {95% CI} n = 20	S (Min–Max) {95% CI} n = 10	A (Min–Max) {95% CI} n = 10	AS (Min–Max) {95% CI} n = 15	*p*
Age (years)	41 (25–62) {6–48}	40 (23–50) {35–47}	47 (37–65) {35–60}	46 (31–64) {41–54}	0.115
BMI	26 (23–31) {25–27}	28 (22–32) {25–29}	24 (22–31) {21–30}	24 (19–33) {11–19}	0.878
Alcohol dependence (years)	0	0	7 (1–25) {1–24}	25 (6–40) {16–29}	0.059
Last alcohol consumption (days)	0	0	22 (3–360) {3–376}	15 (3–700) {3–213}	0.770
Daily amount of the last alcohol consumption (g)	0	0	200 (100–300) {112–287}	200 (100–500) {190–326}	0.387
Nicotine dependence (years)	0	10 (3–20) {5–16}	0	25 (14–40) {21–32}	**0.000**
Daily number of smoked cigarettes	0	10 (1–20) {3–17}	0	20 (15–30) {18–23}	**0.006**
Salivary pH	7.4 (6.4–7.9) {7.1–7.5}	7.2 (5.6–7.8) {6.5–7.6}	7.5 (7.3–8.1) {7.2–8.0}	6.8 (5.9–8.4) {6.4–7.2}	**0.024**
Salivary protein content (mg/mL)	2416 (1282–4998) {2144–3467}	2643 (2276–5323) {2377–3871}	1489 (907–3362) {1267–5107}	2731 (404–5242) {1807–3446}	0.474
CDT (ng/mL)	219 (8–1005) {166–452}	137 (5–640) {78–375}	174 (66–418) {39–370}	122 (1–405) {61–176}	0.351
CDT (ng/mg of protein)	0.103 (0.004–0.783) {0.141–0.043}	0.053 (0.022–0.255) {0.026–0.142}	0.090 (0.051–0.253) {0.130–0.399}	0.039 (0.0009–1.002) {0.016–0.278}	0.420

Data are median values (min-max) {Confidence Interval, CI}. Comparisons between parameters in C, S, A, and AS groups were made using the Kruskal–Wallis analysis, and differences were measured using the Mann–Whitney “U” test. Boldfaced: *p* < 0.05.

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
