# Peer review of "Salivary Carbohydrate-Deficient Transferrin in Alcohol- and Nicotine-Dependent Males"

_jcm, 2020, doi:10.3390/jcm9124054_

Round 1
Reviewer 1 Report
Waszkiewicz et al. Salivary Carbohydrate Deficient Transferrin in Alcohol and Nicotine Dependent Males
Seldom, if any, studies have published on the utility of saliva measurements of CDT upon alcohol intake. The manuscript would be of high interest if the methodology could be improved as explained in the comments below:
INTRODUCTION
Lines 48-49: this sentence is not correct. CDT glycoforms can be asialo-, monoasialo-, disialo-, trisialo-, tetrasialo-, pentasialo- and even hexasialotransferrin (for review, see Helander et al, 2017).
There are also other direct biomarkers of alcohol consumption aside from CDT that could be analyzed, such as phosphatidylethanol (PEth) and Gamma-glutamyl transferase (GGT). There is a wealth of literature that talks about all of the direct biomarkers of alcohol consumption.
Lines 57-59: Elaborate on the role of cigarette smoking CDT results from the literature.
Talk about the influence of age and BMI in CDT levels in reference to previous studies.
METHODS
Participants
State cutoffs of ethanol and cigarettes for each group of participants. Also, state if the participants were hospitalized or outpatients.
Lines 68-69: How were the patterns of drinking and smoking determine in all the groups to know they were honest about their reporting? Were the participants paid to donate the saliva samples?
Table 1
Line 75: Table legend states that data is given in median (95% CI), but in the paragraph for Statistical Analysis, line 104, it says “Descriptive characteristics are presented as means ±S.D.”
Need ranges (minimum and maximum) of each parameter for each group.
The WHO/ISBRA Study on State and Trait Markers of Alcohol Use and Dependence (Conigrave et al, 2002) recommends the use of more than one biomarker of alcohol intake. Also, it has been established that the best method to measure %CDT is HPLC with UV detection (Helander et al, 2017). If the authors do not have the equipment on hand, there is a wide range of laboratories that would measure %CDT with HPL-UV upon receiving the samples. The ELISA method was used many years ago and have been abandoned. In addition, there are no detailed explanations of the kit specifics, such as what glycoform the antibody detects and whether the antibody is monoclonocal or polyclonal; more details would be needed.
The method to measure CDT should be reported as % disialotransferrin to total transferrin and units given in U/L.
A set of reference samples with added amounts of disialotransferrin to total transferrin should be used as calibrators/controls.
Also, it would advisable to measure serum/blood samples in parallel as a control.
Statistical analysis
Data results should be transformed into logarithms to use normal distribution, which is more powerful and then, data should be analyzed for differences with parametric tests. Multivariate linear regression and correlation should be used to analyze BMI, age, GGT and %CDT.
Additional references for the authors:
Conigrave KM, Degenhardt LJ, Whitfield JB, Saunders JB, Helander A, Tabakoff B; WHO/ISBRA Study Group. CDT, GGT, and AST as markers of alcohol use: the WHO/ISBRA collaborative project. Alcohol Clin Exp Res. 2002 Mar;26(3):332-9. PMID: 11923585.
Helander A, Wielders J, Anton R, Arndt T, Bianchi V, Deenmamode J, Jeppsson JO, Whitfield JB, Weykamp C, Schellenberg F; International Federation of Clinical Chemistry and Laboratory Medicine Working Group on Standardisation of Carbohydrate-Deficient Transferrin (IFCC WG-CDT). Reprint of Standardisation and use of the alcohol biomarker carbohydrate-deficient transferrin (CDT). Clin Chim Acta. 2017 Apr;467:15-20. doi: 10.1016/j.cca.2017.03.018. Epub 2017 Mar 18. PMID: 28322729.
Author Response
Seldom, if any, studies have published on the utility of saliva measurements of CDT upon alcohol intake. The manuscript would be of high interest if the methodology could be improved as explained in the comments below:
INTRODUCTION
Lines 48-49: this sentence is not correct. CDT glycoforms can be asialo-, monoasialo-, disialo-, trisialo-, tetrasialo-, pentasialo- and even hexasialotransferrin (for review, see Helander et al, 2017).
-Corrected.
There are also other direct biomarkers of alcohol consumption aside from CDT that could be analyzed, such as phosphatidylethanol (PEth) and Gamma-glutamyl transferase (GGT). There is a wealth of literature that talks about all of the direct biomarkers of alcohol consumption.
-The diagnosis of chronic alcohol consumption was based on the clinical symptoms (ICD-10). The aminotrensferases and GGTP were determined by the routine method in the Hospital in Choroszcz but only in alcohol dependent groups. Unfortunately, we did not determine any other marker from the C and S groups. We, unfortunately, also have no more saliva to determine PEth or GGTP.
Lines 57-59: Elaborate on the role of cigarette smoking CDT results from the literature.
-discussed in the limitations section.
Talk about the influence of age and BMI in CDT levels in reference to previous studies.
-discussed in the limitations section.
METHODS
Participants
State cutoffs of ethanol and cigarettes for each group of participants. Also, state if the participants were hospitalized or outpatients.
- Healthy social-drinkers (C) and chronic cigarette-smokers (S) were outpatients; alcohol-dependent non-smokers (A) and alcohol-dependent smokers (AS) were inpatients. Groups -A and C involved volunteers that were recruited from the Choroszcz Hospital Staff. Now the data about alcohol drinking and smoking are updated in Methodology and Table 1..
Lines 68-69: How were the patterns of drinking and smoking determine in all the groups to know they were honest about their reporting? Were the participants paid to donate the saliva samples?
- The number of cigarettes smoked per day and the amount of daily alcohol drunk were determined on the basis of interviews with the patients/volunteers. Participants were not paid to donate saliva samples. We added this limitation to the limitation section and methodology.
Table 1
Line 75: Table legend states that data is given in median (95% CI), but in the paragraph for Statistical Analysis, line 104, it says “Descriptive characteristics are presented as means ±S.D.”
-Now is corrected in 104 line to median (95% CI).
Need ranges (minimum and maximum) of each parameter for each group.
-CI and min-max of each parameter for each group seems to be too many parameters in one table. We hope we can leave as it stands (median (95%)).
The WHO/ISBRA Study on State and Trait Markers of Alcohol Use and Dependence (Conigrave et al, 2002) recommends the use of more than one biomarker of alcohol intake. Also, it has been established that the best method to measure %CDT is HPLC with UV detection (Helander et al, 2017). If the authors do not have the equipment on hand, there is a wide range of laboratories that would measure %CDT with HPL-UV upon receiving the samples. The ELISA method was used many years ago and have been abandoned. In addition, there are no detailed explanations of the kit specifics, such as what glycoform the antibody detects and whether the antibody is monoclonocal or polyclonal; more details would be needed.
- We, unfortunately, have no more saliva to determine %CDT. We will plan the next study to find it by HPLC. We hope that salivary CDT by ELISA, even if gives negative resuts may add important point to the discussion about CDT usefulness in alcohol dependent persons. We pasted scan of CDT user manual. In the kit we couldn’t find informations what glycoform the antibody detects and whether the antibody is monoclonocal or polyclonal.
-
The method to measure CDT should be reported as % disialotransferrin to total transferrin and units given in U/L.
-This ELISA method gives no opportunity to count % disialotransferrin to total transferrin.
A set of reference samples with added amounts of disialotransferrin to total transferrin should be used as calibrators/controls.
-As above this ELISA method gives no opportunity to count amounts of disialotransferrin to total transferrin used as calibrators/controls.
Also, it would advisable to measure serum/blood samples in parallel as a control.
- We, unfortunately, have no more material from these groups to do additional determinations.
Statistical analysis
Data results should be transformed into logarithms to use normal distribution, which is more powerful and then, data should be analyzed for differences with parametric tests. Multivariate linear regression and correlation should be used to analyze BMI, age, GGT and %CDT.
-Unfortunately, we cannot do logarithms for %CDT analyzes as we have no more material from these groups to do %CDT. However, we analyzed the ELISA CDT data with logarithmic transformation in CDT ELISA as well as with BMI and age. Log data with parametric ANOVA for salivary CDT gave no differences (p=0.449). We found also no significant correlations (log transformed) between CDT -BMI and age. As we had limited time for revision and we found no significant changes in log results in comparison to the text before the revision, we did not decide to change whole text and figures and tables to show it. We sincerely hope that it can stand as we left.
Additional references for the authors:
Conigrave KM, Degenhardt LJ, Whitfield JB, Saunders JB, Helander A, Tabakoff B; WHO/ISBRA Study Group. CDT, GGT, and AST as markers of alcohol use: the WHO/ISBRA collaborative project. Alcohol Clin Exp Res. 2002 Mar;26(3):332-9. PMID: 11923585.
Helander A, Wielders J, Anton R, Arndt T, Bianchi V, Deenmamode J, Jeppsson JO, Whitfield JB, Weykamp C, Schellenberg F; International Federation of Clinical Chemistry and Laboratory Medicine Working Group on Standardisation of Carbohydrate-Deficient Transferrin (IFCC WG-CDT). Reprint of Standardisation and use of the alcohol biomarker carbohydrate-deficient transferrin (CDT). Clin Chim Acta. 2017 Apr;467:15-20. doi: 10.1016/j.cca.2017.03.018. Epub 2017 Mar 18. PMID: 28322729.
-We incorporated these references into text.

Reviewer 2 Report
This manuscript describes measurement of CDT in saliva by Enzyme-Linked-Immunosorbent-Assay in controls (C), chronic cigarette smokers (S), alcohol-dependent non-smokers (A), and alcohol-dependent smokers (AS). While the paper is interesting, there are some concerns which are described below.
There is not enough information regarding the C and S. For example, where and how were they recruited - from the community or the lab, etc?
There is no information on the “last alcohol consumption (days)” and the “daily amount of the last alcohol consumption (g)” for C and S. Therefore, the comparison between CDT levels in the control groups and the alcohol dependent groups are not valid.
For all groups, more information is needed about their drinking, not just last day and daily amount of last day. CDT is a marker for chronic drinking, as noted by the authors CDT appears “in serum after regular high alcohol intake of more than 50–80 g of alcohol per day for at least 1 week.” Thus, drinking over at least the last week (preferably longer, like the past month) of all participants (C, S, A, and AS) should be included and include numbers of drinking days and number of heavy drinking days during a time period.
The conclusion that “Therefore, the salivary CDT can not be used as alcohol dependence marker as is it does in serum” is overstated, not only because of the lack of alcohol consumption data (mentioned above), but the CDT assay. The preferred method for measuring CDT in SERUM is HPLC/UV method (Helander et al., 2017). Thus, comparing salivary CDT measured using Enzyme-Linked-Immunosorbent-Assay to serum CDT measured using HPLC/UV is not a valid comparison. If at all, the conclusion should be modified to “Therefore, the salivary CDT cannot be used as alcohol dependence marker as measured by Enzyme-Linked-Immunosorbent-Assay”.
Finally, the authors should check throughout the manuscript for grammar and missing words.
Author Response
This manuscript describes measurement of CDT in saliva by Enzyme-Linked-Immunosorbent-Assay in controls (C), chronic cigarette smokers (S), alcohol-dependent non-smokers (A), and alcohol-dependent smokers (AS). While the paper is interesting, there are some concerns which are described below.
There is not enough information regarding the C and S. For example, where and how were they recruited - from the community or the lab, etc?
-Groups -S (chronic cigarette-smokers, social-alcohol-drinkers) and C (healthy social-drinkers) involved volunteers that were recruited from the Choroszcz Hospital Staff. We added this information in the methodology section.
There is no information on the “last alcohol consumption (days)” and the “daily amount of the last alcohol consumption (g)” for C and S. Therefore, the comparison between CDT levels in the control groups and the alcohol dependent groups are not valid.
-As C and S groups had not consumed alcohol for 10 days before study recruitment we still involved them as social drinkers, but ”0” alcohol data was provided for last alcohol consumption (days)” and the “daily amount of the last alcohol consumption (g). We described it in the methodology section and corrected in Table 1.
For all groups, more information is needed about their drinking, not just last day and daily amount of last day. CDT is a marker for chronic drinking, as noted by the authors CDT appears “in serum after regular high alcohol intake of more than 50–80 g of alcohol per day for at least 1 week.” Thus, drinking over at least the last week (preferably longer, like the past month) of all participants (C, S, A, and AS) should be included and include numbers of drinking days and number of heavy drinking days during a time period.
-We included all the data we had about alcohol drinking in all groups. Now it is corrected in Methodology and Table 1.
The conclusion that “Therefore, the salivary CDT can not be used as alcohol dependence marker as is it does in serum” is overstated, not only because of the lack of alcohol consumption data (mentioned above), but the CDT assay. The preferred method for measuring CDT in SERUM is HPLC/UV method (Helander et al., 2017). Thus, comparing salivary CDT measured using Enzyme-Linked-Immunosorbent-Assay to serum CDT measured using HPLC/UV is not a valid comparison. If at all, the conclusion should be modified to “Therefore, the salivary CDT cannot be used as alcohol dependence marker as measured by Enzyme-Linked-Immunosorbent-Assay”.
-We changed this conclusion as suggested by the Reviewer.
Finally, the authors should check throughout the manuscript for grammar and missing words.
-We corrected the manuscript for grammar and missing words.

Round 2
Reviewer 1 Report
Thank you to the authors for their responses.
Questions/Comments:
- METHODS, Participants.
- What type of assessment methods were used to collect the self-reporting alcohol consumption?
- What were the alcohol cutoffs (drinks per day, days of drinking) for the groups (C), (A) and (AS)?
- It is not acceptable to say that there is too much information for one table. The confidence intervals are an estimation of the range that contains the true mean of the population being sampled. The minimum and maximum values are important because it gives a true scatter value of the samples being analyzed, therefore it is important to state these values unless all of the individual values are provided. If the minimum and maximum values do not fit in the table, write those values in the corresponding text.
- Negative, and positive, results are interesting if the experimental design and methodology are properly carried. The problem is that the methods employed in this study have too many pitfalls; and some of them could have been avoided at the time of the experiment. For instance, when using a commercial kit, it is advisable to gather the most information either from the website or by directly calling the vendor before the purchase. If that information is not available, then it is better to select another kit.
- Many scientists have published data from experiments performed in a distant past with samples long gone. This is understandable and publishable if it affects very specific points that could be addressed in the manuscript. However, in this case, it affects the whole body of work presented in this manuscript.
- The authors state that they are planning to analyze another set of saliva samples by HPLC. This manuscript would be much more powerful after these additional saliva samples are analyzed and results obtained.
Author Response
1. METHODS, Participants.
- What type of assessment methods were used to collect the self-reporting alcohol consumption?
- The number of cigarettes smoked per day, years of nicotine dependence, as well as the amount of days and grams of daily alcohol drunk and years of alcohol dependence were determined on the basis of interviews with the patients/volunteers and conducted by a professional psychiatrist with extensive clinical experience. We stressed it in the methodology.
- What were the alcohol cutoffs (drinks per day, days of drinking) for the groups (C), (A) and (AS)?
-The cut-offs for chronic alcohol drinking (in alcohol dependent persons -A and AS groups) were: at least 3 days of continuous alcohol drinking of at least 100g of pure ethanol per day. Participants from the A and AS groups were suffering from alcohol withdrawal state. The alcohol cut-offs for social alcohol drinking (C and S groups) were: drinking not more than 20-30g of pure ethanol per day, these volunteers had not consumed alcohol for 10 days before the study recruitment and ”0” alcohol data was provided for the last alcohol consumption (days) and the daily amount of the last alcohol consumption (g). We added this information to the methodology section.
2. It is not acceptable to say that there is too much information for one table. The confidence intervals are an estimation of the range that contains the true mean of the population being sampled. The minimum and maximum values are important because it gives a true scatter value of the samples being analyzed, therefore it is important to state these values unless all of the individual values are provided. If the minimum and maximum values do not fit in the table, write those values in the corresponding text.
-We incorporated min-max values into Table 1.
3. Negative, and positive, results are interesting if the experimental design and methodology are properly carried. The problem is that the methods employed in this study have too many pitfalls; and some of them could have been avoided at the time of the experiment. For instance, when using a commercial kit, it is advisable to gather the most information either from the website or by directly calling the vendor before the purchase. If that information is not available, then it is better to select another kit.
-The sensitivity of the assay is 9.38 ng/mL and the detection range is 15.63-1000 ng/mL. This kit recognizes Human CDT in samples. No Significant cross-reactivity or interference between Human CDT and analogues was observed.
We added it to the 2.3.2. Analytical methods subsection.
4. Many scientists have published data from experiments performed in a distant past with samples long gone. This is understandable and publishable if it affects very specific points that could be addressed in the manuscript. However, in this case, it affects the whole body of work presented in this manuscript.
-We realize that more powerful CDT assays as HPLC can give more accurate results. However, ELISA tests can be performed even in not very specialized laboratories. Our pilot study led by Kratz at al [17] tried to identify some of salivary glycoproteins, altered by chronic alcohol consumption. By using defined molecular weight of glycoproteins and glycosidic bonds, we identified salivary glycoproteins with significant alcohol-induced differences in glycosylation (α2,3-sialylation, fucosylation, and expression of T-antigen). These salivary glycoproteins, changed by the chronic alcohol use, included α-amylase, clusterin, haptoglobin, light and heavy chain immunoglobulin, and transferrin. Therefore, trensferrin was suggested as a potential marker for diagnosis of chronic alcohol drinking. Therefore, we expected to find salivary CDT as a good marker of chronic alcohol drinking (alcohol dependence syndrome). So, we decided to check salivary CDT with a simple methodology. We highlighted it in the diccussion and in the Limitations section.
5. The authors state that they are planning to analyze another set of saliva samples by HPLC. This manuscript would be much more powerful after these additional saliva samples are analyzed and results obtained.
- The salivary samples for our study were collected for over a year and a half. To do HPLC study we need similar time. We sincerely hope that our simple ELISA results, if published, can give helpful information for researchers who check noninvasive alcohol markers and suggest other methods for these findings.

Reviewer 2 Report
The authors addressed all my concerns.
Author Response
The authors addressed all my concerns.
-Thank You.
